# Self-Assembly of DNA-Grafted Colloids: A Review of Challenges

**DOI:** 10.3390/mi13071102

**Published:** 2022-07-14

**Authors:** Manish Dwivedi, Swarn Lata Singh, Atul S. Bharadwaj, Vimal Kishore, Ajay Vikram Singh

**Affiliations:** 1Department of Physics, Banaras Hindu University, Varanasi 221005, UP, India; dwdmanish19@gmail.com (M.D.); vimalk@bhu.ac.in (V.K.); 2Department of Physics, Mahila Mahavidyalaya (MMV), Banaras Hindu University, Varanasi 221005, UP, India; 3Department of Physics, CMP Degree College, University of Allahabad, Prayagraj 211002, UP, India; atulsbharadwaj@gmail.com; 4Department of Chemical and Product Safety, German Federal Institute of Risk Assessment (BfR), Maxdohrnstrasse 8-10, 10589 Berlin, Germany

**Keywords:** self-assembly, DNA-grafted colloids, kinetic arrest

## Abstract

DNA-mediated self-assembly of colloids has emerged as a powerful tool to assemble the materials of prescribed structure and properties. The uniqueness of the approach lies in the sequence-specific, thermo-reversible hybridization of the DNA-strands based on Watson–Crick base pairing. Grafting particles with DNA strands, thus, results into building blocks that are fully programmable, and can, in principle, be assembled into any desired structure. There are, however, impediments that hinder the DNA-grafted particles from realizing their full potential, as building blocks, for programmable self-assembly. In this short review, we focus on these challenges and highlight the research around tackling these challenges.

## 1. Introduction

Self assembly-a process crucial to life-plays an important role in the formation of complex biological structures. From the formation of cell membranes and protein aggregates to the folding of nucleic acid into functional conformation, the process is ubiquitous in biological functions [1,2,3]. Self assembled structures form when individual units arrange themselves spontaneously, into an ordered structure, as a result of specific local interactions. This biological designing principal has been extensively explored by researchers to produce materials with tailor made properties [4,5,6,7,8,9]. The approach involves the self assembly of individual units; these units contain information about the interactions, and the assembly pathways that lead to the final configuration. Programmable assembly has emerged as a promising bottom up approach for producing materials with desired physical and chemical properties. Such materials can be very beneficial in number of applications ranging from bio-medicine [10,11,12,13,14], sensing [15], plasmonic [16], and photonics [17] to microelectronics [18].

Programming the assembly of a desired structure thus has two crucial components; the building blocks or the individual units, and the specific local interactions encoded within these units. Although there are several methods of creating programmable blocks, when it comes to encoding information about the local interactions, DNA makes a very good candidate due to its highly directional and sequence specific binding properties [19]. DNA is a biomolecule, which stores the genetic information in four bases of nucleic acid, known as A, T, C, and G. The sequence of these four bases- which are arranged on a phosphate backbone- determines the structural function of the DNA. Double helical DNA or more complex molecules are formed when these bases pair-up following Watson Crick base pairing which says that A can pair only with T and C can pair only with G [20]. This specific pairing interaction, together with the possibility to generate a DNA strand of desired sequence and length, is the key to a very precise and programmable assembly.

Two main routes are taken for achieving self-assembled structures using DNA. First is being the DNA origami, which uses strands of DNA as the building blocks to create complex structures through DNA hybridization. The origami involves a very long single strand of DNA, which then is folded and held in desired configuration using the short DNA strands as staple [21,22,23]. Though posing great promise for applications such as molecular robotics [24,25], biosensing [26], bioimaging [27] and drug delivery [28], this technique comes with its own challenges. Issues such as size limit, stability, and cost of production at large scales, on top of the the fact that DNA comes with its own (not so remarkable) material properties, limit the applications of DNA origami [6,29]. The other approach, which addresses the above mentioned issues to some extent, is to create building blocks (of some synthetic material) with DNA as binding sites. One such example is of colloids grafted with single strands of DNA; a system that has attracted much interest recently [29,30,31,32,33].

DNA grafted particles assemble through the hybridization of the grafts, under favourable conditions. In principle, any desired morphology can be achieved by tuning the sequence of the bases on the grafts. Some other obvious parameters are the density and the size of grafts. In practice, however, it still remains challenging to generate a vast variety of structures using DNA grafted particles. The reasons being (i) inability to code sufficient information required to direct the spontaneous formation of any arbitrarily complex structure, structures produced so far have been of very limited morphology [19], and (ii) formation of kinetically arrested structures. A subset of the problem is a steep transition of building blocks from dispersed to aggregated phase, as a function of temperature. This sharp transition also contributes to the kinetic arrest [29,32,33]. In this review, we will focus on these challenges and cite the studies focused on resolving these issues.

Although, in this review, we focus on the self assembly of DNA grafted solid particles, recently, DNA coated droplets have also emerged as a model system for sequential self assembly of the building blocks due to their natural patchiness. The patchiness results from the fact that the linkers can diffuse on the surface of the droplets, and can be recruited into patches, giving rise to defined valency. We cite some relevant literature for the readers interested in DNA coated droplets [34,35,36,37,38,39,40]. It is also worth mentioning here that, as a strategy, self-assembly is largely confined to nano/micro size components. However, it is equally feasible for bigger building blocks to self-assemble under favourable conditions [41].

The structure of this review article is as follows: in the Section 2, we give a brief introduction of DNA grafted particles as a system, we analyze the building block and briefly discuss the control parameters that can be used to tune the effective interparticle interactions. Next, we give a detailed account of the challenges faced by the community in engineering programmed materials using DNA grafted particles. We also go through the recent studies about ways to tackle the hurdles that limit the possibilities of what can be achieved using this bottom up approach. We end the article with a summary of the future goals to be achieved, in order to realize the full potential of DNA-grafted building blocks.

## 2. System

### Building Block

Building blocks of DNA mediated self-assembly includes a core, made up of particle with size in approximate range of $10^{-10}$ to $10^{-6}$ meters, and a shell that consists of DNA molecules attached to the surface of the core particle, see Figure 1a. These DNA strands have one end pinned to the surface of core and the other end remains free to interact with other building blocks. The free ends of DNA strands are the unpaired bases which are involved in the hybridization process and hence, appropriately called as sticky ends as shown in Figure 1b. These sticky ends are encoded with the sequence specific information about the interactions and the assembly pathways to facilitate the formation of desired assemblies.

The sequences on the sticky ends decide the interactions between the neighboring building blocks, and the grafts attached to the core particles control the accessibility of the sticky ends belonging to neighboring blocks. Locally, only those connections are formed and preserved, which are thermodynamically favored. The desired macroscopic structural ordering, achieved by tuning the interactions, between the building blocks, can exhibit a higher degree of order having the translational and/or rotational symmetries, as shown in Figure 1c.

#### Control Parameters

As discussed, the building blocks are made up of a core particle and DNA strands that are grafted on the core particle. The core particle could be of any composition: metal, semiconductors, or polymers. The final product depends on the type and size of the core particle [42]. Additionally, the interactions between building blocks could be modulated using the parameters associated with the strands. These strands are usually composed of three regions: a recognition sequence that binds with the core particle, a double helical DNA-strand that acts as a spacer, and at the end there is a “sticky end”. It is the sticky end that hybridizes with the sticky end of the strands belonging to other particles, and thus, binds them together (see Figure 1b). Sometimes, to increase the flexibility of the grafts, there are inclusions of flexible regions (of unpaired bases) either near the surface of the particle, or close to the sticky ends. The sequence of the sticky end determines the specific interactions whereas, the length of the sticky end determines how fast the transient hybridization/dehybridization of duplexes, formed by these sticky ends during the assembly process, will take place [29]. Apart from that, overall length of the strand is another parameter that is used to control the porosity of the final structure. Having longer lengths, however, makes it more difficult for the particles to diffuse and re-arrange [43,44]. Another important control parameter is the areal density of the grafts, increasing grafting density increases the effective interparticle interactions [43,45].

Flexibility of the strands increases the volume accessible to the sticky ends, enabling better sampling to find stable interactions during the assembly process [42]. Flexible strands also diffuse more easily. Anisotropy/asymmetry in the building blocks can be utilized to generate a rich phase diagram [5]. Effect of these parameters will be discuss in more detail, in the next section and more references could be found there. Another parameter, that affects the assembly, is the size ratio of the particles in case of a mixture of the building blocks [46]. This size ratio also controls the aspect ratio of the final product. Salt concentration is another parameter that affects the effective interactions in the system. Changing the salt concentration can change the final structure [47]. To find out more about the effect of these parameters, kindly refer to [33].

## 3. Challenges

DNA grafted particles hold great promise as a tool to create programmed structures [6,29,32,48]. Ever since the first demonstration of the assembly of DNA-grafted nanoparticles into crystalline structure, by Mirkin [30] and Alivisatos [31], there has been extensive research around the possibilities offered by the self-assembly of DNA-grafted particles [32,33], as well as to develop a microscopic understanding of various mechanisms involved [49,50,51,52,53,54]. In principle, any target structures can be created using these building blocks, if the particles are functionalized with complementary strands of user defined sequences. However, assembling a breadth of complex structures has remained elusive so far. In the following, we give an account of the challenges that have to be worked on, in order to be able to use DNA-grafted particles to generate a wide spectrum of structures with desired properties. We will also mention the studies that propose a solution of these existing problems.

### 3.1. Encoding Interactions

The most common approach for assembling DNA-grafted particles is via direct hybridization of the grafted sequences to each other, following the Watson–Crick base pairing. Although the inter-particle interactions can be fine-tuned by varying the sequence of base pairs on the strands, isotropically-functionalized spherical particles that assemble via direct hybridization of the grafts only lead to limited crystalline structures, mostly face centered or body centered phases depending upon whether system is one-component or binary in terms of the building blocks [33,55,56,57]. Though additional parameters such as the sequence, length, and density of the grafts can have an effect on the lattice parameters as well as the crystallographic symmetries, still the resulting structures remain limited to the closed-packed crystalline symmetries [58,59,60]. To generate more complex structures, or structures that lack periodicity and/or isotropy while using isotropic building blocks, one would need: (i) the building blocks that differ from each-other, and (ii) the local interactions that favor the desired structure. This, eventually, would require specifying a large number of distinct specific interactions, even for designing a small assembly [33,61].

Another way to achieve a richer phase space is by introducing anisotropy in the system [5]. This, in present context, can be achieved either by functionalizing each particle with more than one kind of DNA-strands, or, by using the shape of the colloids. Here, the “kind of strand" stands for the sequence that makes up the strand. Coating particles with strands of more than one kind brings the possibility of generating more complex structures; however, there is a substantial entropic cost involved. Hybridization of strands, in such a situation, demands the two particles (that are supposed to be neighbors) to orient themselves with respect to each other in a way that the complementary strands can see each-other. On the other hand, grafting particles with one kind of sequence maximizes the probability of hybridization, and hence the stability. Moreover, there is an upper limit on how many of “unique interactions" can be modeled using DNA-strands. This limitation stems from the fact that there are possibilities of unwanted hybridization between partially complementary strands that belong to the same particle, as well as the possibilities of formation of secondary structures such as hairpin, which eventually results in loss of binding sites [62,63]. It is also challenging to synthesize a colloid grafted with different sequences which contributes to relatively sparse studies involving such building blocks [9,64].

An alternate way of creating anisotropic interactions while using spherical colloids is to create DNA-functionalized Janus particles [9,64,65,66,67,68,69]. It has been found that these Janus particles self assemble into structures that are more complex and diverse than the closed packed crystals. The anisotropy due to the shape of the core particle has been exploited too [70,71,72]. These studies involved DNA-decorated nanorods [73], DNA-modified triangular bipyramids [74], DNA functionalized polyhedral mixed with DNA functionalized spheres [75], DNA-shelled nano cubes [76], and anisotropic nanoparticles of different shapes assembled using DNA ligands [77]. Recent in the line is to break the symmetry, of a system of isotropic building blocks, by introducing smaller isotropic building blocks in the system. These smaller particles are functionalized with strands that are complementary to the strands grafted on the bigger particles, and the grafting density is also lower in comparison to that of the bigger particles [78,79]. The strategy of using DNA origami, to assemble particle, opens up the possibilities that have not been realized otherwise. DNA origami is utilized in two ways: (i) as cages to host the particles, the cages then bind through DNA strands [80,81], and (ii) as anisotropic binding mediator to create complex structures [21,22,23,70]. DNA origami could also be used as mesh frame to assemble nanoparticles into materials of desired morphology [22,23,70,82,83,84,85].

The other approach involves the use of free DNA strands (linkers), dispersed in the medium, to encode the information of pair interactions. The simplest approach involves coating the particles with strands that are not complementary to each other, so that there is no direct hybridization, and the binding is actually mediated through the linkers. The linkers usually have two regions of sequences that are complementary to the sequence of the grafts. The two regions of linker bind to the complementary strands belonging to two different particles, eventually binding them together [44,56,57]. Figure 2, reproduced from [44], shows the scheme for the simplest approach for the linker mediated binding. In this paper, which is one of the early studies introducing the idea, the authors have experimentally investigated the linker mediated assembly of DNA-grafted nanoparticles, as a function of linker length and the number of linkers per particle. Figure 3a shows the order in the system for a given linker length, and Figure 3b shows the variation of the order as a function of linker length. Figure 3c show the dependence of the interparticle separation on the linker length.

Using linkers, its possible to encode large number of specific interactions with fewer unique sequences in comparison to the systems where the assembly is achieved via direct hybridization. By changing the concentration, length, sequence and symmetry of linkers, the interparticle interactions and the binding kinetics could be tuned. The complexity can be increased by using a cocktail of linker sequences which effectively amounts to increasing the number of unique interactions. These linkers thus emerge as a useful tool to program more complex and diverse structures [61,63,86,87,88]. Apart from gaining a control over final output, adding information via linkers also enriches the phase diagram with features such as re−entrant melting and broader temperature range within which the melting takes place [89,90]. These studies, however, involve the grafting of particles such that the particles could bind through direct hybridization, and the linkers act as displacing strands. The sequence of these linkers is chosen such that these linkers are complementary to the subunits of the grafted strands, and under suitable conditions, these linkers react to the bridges that bind two particles and replace one of the strands. This process is called strand displacement reaction and results into non-bridging duplexes. Free DNA strands could even be introduced in the system post assembly, to reprogram the interactions and achieve structural transitions from one to another phase [87,91].

### 3.2. Kinetic Arrest

Another grand challenge, while equilibrating the system towards the prescribed structure, is the kinetic arrest of the system into some random aggregate that fails to anneal further. Except for the cases when the particles are small (smaller than few hundred nanometers), the DNA-grafted particles, generally, find it difficult to re-arrange while they are bound to each other [29,32,45,92]. Many factors have been held responsible for this kinetic arrest: the grafting density, length and the rigidity of the grafts, inhomogeneity in distribution of the grafts, and the roughness of the particle’s surface [92,93,94]. Another reason of kinetic arrest lies in the sharp melting of the DNA strands that we will discuss in detail in the next subsection (see Figure 4). Weak interactions are found to help in overcoming the kinetic barriers [95]. One of the strategies involved is to modulate the interaction between building blocks by changing the length and/or, the sequence of the sticky ends. Having long sticky ends slows down the dehybridization and hence the re-organization [29,96]. Reducing the length of the sticky ends (or the number of hybridizing bases) decreases the melting temperature, which affects the diffusivity of the particles negatively. A balance, however, could be achieved by increasing the areal density of the grafts, as melting temperature increases with increasing graft density. It is shown that having high density of grafts with short sticky ends, helps in avoiding kinetic traps and drives system towards equilibrium structure [6,97]. Another advantage of having high surface coverage is to compensate for any uneven distribution of DNA strands, or any underlying surface heterogeneities [45,54,97].

In addition, the total length of the strand (including recognition sequence, spacer and sticky ends) is an important parameter to control the porosity of the assembled structure. The studies on DNA-grafted nanoparticles show that the final structure depends on the size of the strands [44,96,98,99]. These studies, however, also show that there is a limited range of the strand length over which the system equilibrates into an ordered structure, choosing any arbitrary length drives system towards kinetic arrest. Mao et al. [43] again proposed that increasing the grafting density could overcome this problem. Another important aspect here is the flexibility of the grafts; adding flexibility to the grafts renders the sticky ends more freedom to explore their neighborhood as well as an access to increased free-volume. Having many concurrent hybridization events makes the rolling and re-arranging easier [55,57,100,101,102]. There is, however, an upper limit of the degree of the flexibility that can be added to the design of the grafts. Going beyond the limit again results into deviations from the ordered structures [42,101,103], the deviations being attributed to the random coiling of the flexible grafts [29].

### 3.3. Sharp Melting Transition

DNA-grafted particles behave like gases at temperatures above melting temperature ($T_{M}$), and condense below $T_{M}$ through duplex formation between single stranded grafts [54,92,104]. This transition takes place in a narrow window of temperature which means particles could just land into some random aggregate below $T_{M}$, without getting enough opportunities to re-arrange into the desired structure. This problem is further aggravated with the increasing size of the colloids. Several studies have modeled the effective interaction between DNA-grafted particles [49,52,53,54,104,105,106], and it is found that the melting transition of DNA-grafted particles is much steeper than the melting of DNA-strands in solution (see Figure 4). Broadly speaking, there are two different approaches taken towards softening of this sharp temperature dependence. First being the fabrication of colloids using mobile DNA strands [93], Muelen et al. prepared these particles such that the DNA-grafts were anchored to a lipid bilayer which coats the particles and is in liquid phase, rendering lateral mobility to the colloids (see Figure 5A–D). This mobility helps particles diffuse, and hence, they associate/dissociate less dramatically leading to a wider temperature range for transition (see Figure 5E). Having a wider window means particles could re-arrange themselves better and improve the chances of reaching to the equilibrium structure. Having mobile strands also helps with any surface heterogeneities and non-uniform distribution of grafts. In this spirit, there have been continuous studies to understand and to enrich the phase diagram of the assembly of the system of particles that are grafted with mobile strands [107,108,109,110].

The other approach is to introduce competing interactions in the system. Mongnetti et al. [111] proposed a two linkage model (scheme shown in Figure 6) in a binary system of colloids. They designed the sequences of the grafts such that there are both the strong and the weak bonds between the grafts belonging to the two species of colloids. They show that as the temperature decreases, there is switching from the stronger to the weaker linkages, due to energetic reasons, resulting into a control over temperature dependence.

This idea was taken further [112], where the weaker bonds are between the strands belonging to the same particle. This results in a re-entrant melting; when the temperature is lowered, first, interparticle bonds (strong) are formed; however, when the temperature is lowered further, there are intraparticle bonds replacing the interparticle bonds as both are mutually exclusive. Another way of introducing competing interactions is by using free DNA linkers in the solution [90] (scheme proposed by Rogers et al. is shown in Figure 7A). As mentioned in the previous section, these strands are designed so that they can react with the double stranded bridges, that bind two particles, and replace one of the strands. This displacement changes the magnitude of hybridization between the DNA grafted particles. The magnitude of hybridization can be controlled by the concentration and sequence of these strands, making it possible to achieve equilibrium structure over a wider range of temperature (see Figure 7B,C). Here, we only reprint the results for one type of displacing strand, the authors also introduced two types of displacing strands in the system, and observed a re-entrant melting that could be tuned by tuning the displacing strands concentration. Strand displacement can also be used to switch between different morphologies, and to achieve multiple melting points [113].

## 4. Summary and Future Direction

The objective of the future research could be divided into two categories: short term goal and long term goal. The short term goal is more about gaining enough understanding of how to design the appropriate building blocks, and, once we have the building blocks how to make sure that the system reaches the target structure. The first relevant step in this regard is to encode sufficient information in the building blocks, without paying too much in entropy costs. Generating complex structures of large size remains a challenge, complex structures require building blocks to be heterogeneous and one way of achieving it is to model the interactions anisotropically. As elaborated in Section 3.1, modeling anisotropic interactions comes with its own limitations as each specific interaction requires an unique sequence of the base pairs; if particles are grafted with strands of more than one kind, hybridization of two sequences that belong to two different particles demands these two particles to be in a certain orientation with respect to each other. There are also possibilities of loosing binding sites due to unwanted hybridizations.

Further efforts are required to achieve the complexity at the level of individual building blocks by either synthesizing highly anisotropic core particles or by using multiple types of strands. These strands can be used as grafts: either grafting each particle with multiple types of strands, or having more than one set of building blocks. Having multiple species, however, requires design rules to program the assembly and simulations can be very useful here [62,114]. Using linkers to encode assembly instructions presents a promising route to create complex structures. It also helps in smoothening the sharp temperature dependence. These linkers could be single strands of DNA [61], smaller particles with specific binding interactions [79], or even DNA origami [33]. DNA origamis can carry a larger number of specific interactions due to their prescribed shape and binding specificities. Moreover, these DNA constructs come with a precise control over their strctural topology and, hence, can guide the self assembly with high accuracy. Recent studies on origami mediated assembly, though mostly involving nanoparticles, have already revealed rich and interesting phase behaviour [115,116,117,118].

As discussed in Section 3.1, origamis could also serve as cages to host nanoparticles. Gang et al. have shown that these origami cages, hosting the nanoparticle, together with other nano-components, co-assemble into diamond structure [80] (see Figure 8 and Figure 9 for the scheme and various structure assembled using different type of origami cages). Combining DNA constructs with DNA-grafted particles thus poses the possibilities of creating highly complex structures with flexibility and accuracy that is not achieved yet. The idea, however, is still in its infancy and needs to be explored further.

The next hurdle in line is the kinetic arrest of the system into disordered structures. Ideally, its possible to achieve any structure by tuning the grafts, practically however, the DNA-grafted colloids often get stuck into random aggregates instead of reaching to the targeted structure. A number of factors, such as the density of grafting, the length of DNA strands, and sharp thermal activation have been credited for this kinetic arrest. The re-arrangement of the particles in the system is essential to reach the equilibrium structure. One of the reasons of kinetic arrest also lies in the fact that the melting transition of DNA strands happens in a very narrow range of temperature. This narrow range, combined with the fact that DNA coated particles generally find it difficult to roll around each other, leads to kinetically trapped system without further re-arrangements towards the desired assembly. One of the immediate goals is to broaden the temperature window in which the association-dissociation takes place. We have given an account of the efforts made so far to increase the temperature window of assembly so that the particles could re-arrange themselves towards the equilibrium in Section 3.3.

As discussed in the Section 3.2, there have been continuing efforts to propose design rules to avoid the kinetic arrest and achieve a successful assembly. There are a number of forces at play; apart from the base pairing, there are steric, Van der Waal’s and electrostatic interactions present in the system. Researchers have tried to experimentally probe the interaction potential in a system of DNA-coated particles, and relate it to various design parameters such as sequence, length and areal density of the grafts [49]. Additionally, there are several theoretical and computational investigations to pinpoint the effect of these system parameters on the dynamics and thermodynamics of the assembly process [50,52,105].However, a complete understanding of the assembly pathways and the effect of these parameters on the assembly dynamics is still to be developed. Using predictive tools and simulations, to gain the conceptual insights, would help in obtaining design elements that could avoid kinetic arrest and lead to successful assemblies. Growing interest in machine learning has further enriched the reverse engineering of the self-assembly, including the field of DNA nano-technology [29,119,120]. We expect the combined experimental-computational efforts to render the understanding, and thus the control, required to design and guide the building blocks into any desired structure

The bigger picture and the long term goal, however, should be about reaching to the point where a generic system of DNA-grafted particles can be used to generate any arbitrary structure. Generating specific building blocks for each target structure is costly and time-consuming as it requires grafting specific sequences onto colloids every time. Linker mediated assemblies could be controlled by changing the concentration or the size of linkers. In such assemblies, the final structure depends upon concentration, size and sequence or symmetry of the linkers dispersed in the solution. This, however, solves the problem only upto a certain extent. There is still a long way to go in order to be able to successfully generate any specific structure from a generic set of building blocks, without getting trapped into a disordered aggregate.

## Figures and Tables

**Figure 1 micromachines-13-01102-f001:**
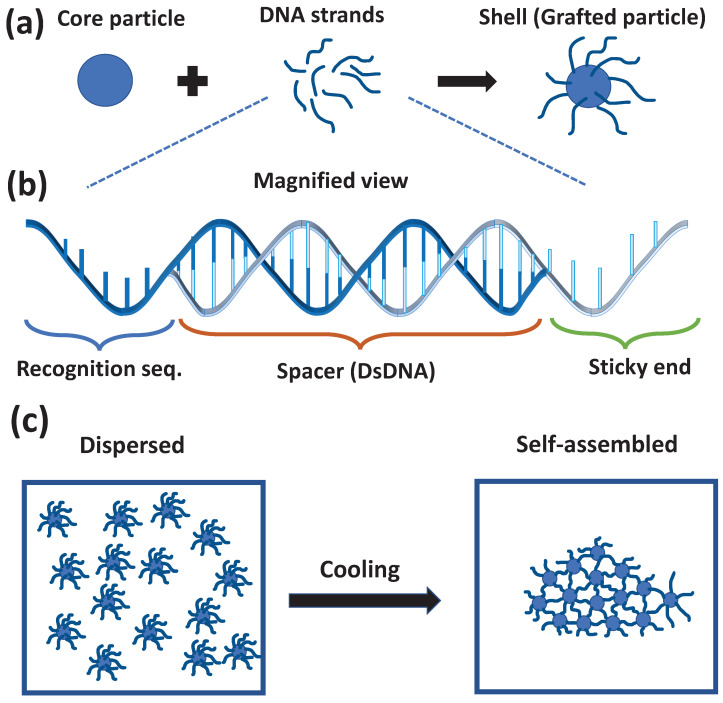
(**a**) is a cartoon representation of a DNA-grafted colloid. (**b**) shows the magnified view of the strands. The most general scheme of designing a DNA strand, for grafting purpose, involves a recognition sequence followed by a spacer and at the end there is sticky end. The recognition sequence is a sequence of unpaired bases that binds to the core particle. Spacer is used the control the overall length of the grafts. The sticky end has unpaired bases that hybridize with the sticky end of other building blocks. (**c**) is a cartoon to depict the assembly of DNA-grafted particles.

**Figure 2 micromachines-13-01102-f002:**
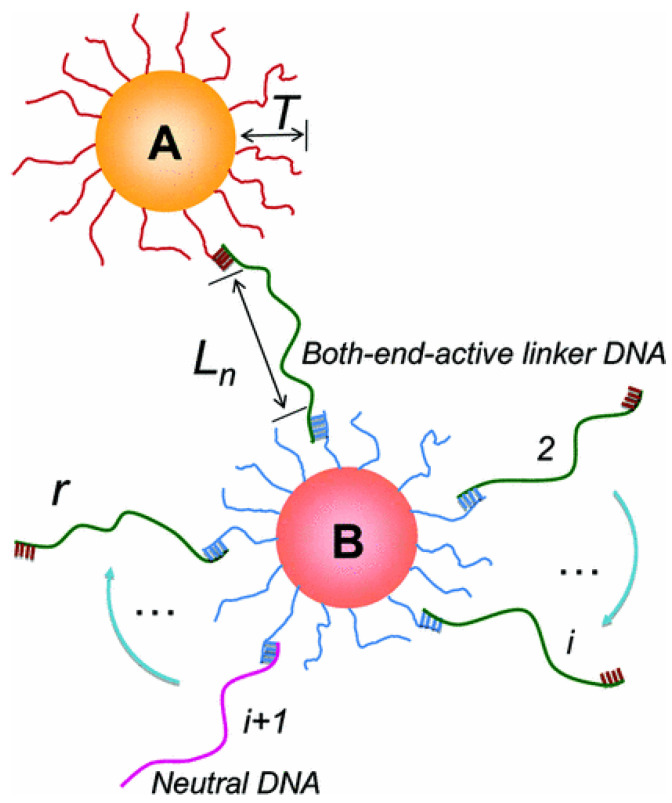
A schematic representation of linker-mediated binding of DNA-grafted particles proposed by Xiong et al. The system, with core particles A and B, has two types of linkers: linkers with both ends active called the active linkers and linkers with just one end active that are called as neutral DNA. The length of the linkers is denoted by $L_{n}$, *r* is the total number of linkers that each particle connects to, via the hybridization of the sticky ends of the grafts. The linkers have 15 base long recognition sequence on both the ends and *n* represents the length of central flexible polythymine. Reprinted with permission from [44] copyright 2009 by American Physical Society.

**Figure 3 micromachines-13-01102-f003:**
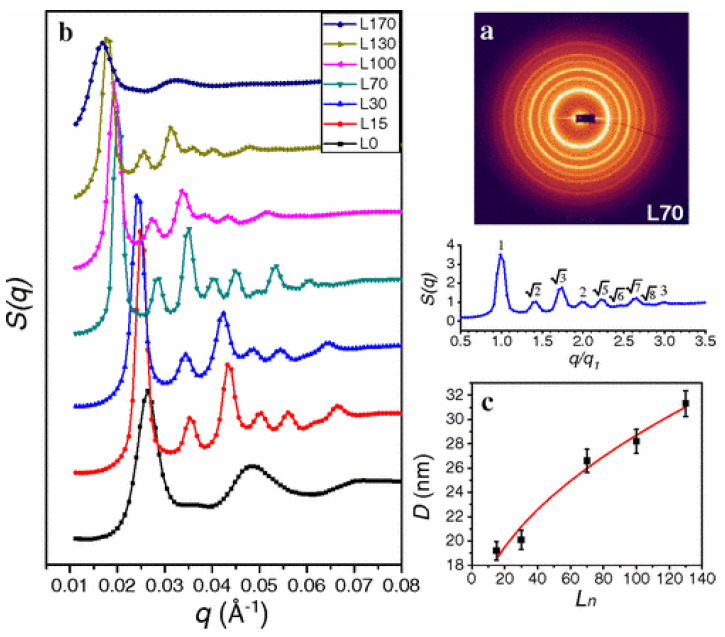
Structure of linker-mediated assemblies in the system shown in Figure 2. (**a**) contains the SAXS pattern and the structure factor for a system with $L_{n}=70$, (**b**) shows the variation of the structure factor with $L_{n}$ for r=36. It can be seen that increasing $L_{n}$ beyond a certain value decreases the order in the system. (**c**) gives the value of interparticle surface to surface distance (*D*) as function of $L_{n}$. Reprinted with permission from [44] copyright 2009 by American Physical Society.

**Figure 4 micromachines-13-01102-f004:**
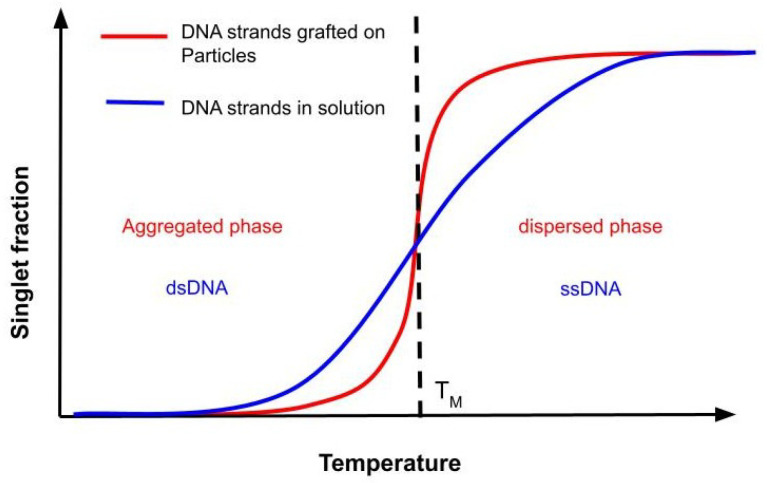
Representative melting profile of DNA-grafted particles (shown in red), and of DNA strands in solution (shown in blue). $T_{M}$ is the temperature at which half of the bases pair up. Below $T_{M}$, DNA grafted particles aggregate and make an assembly whereas, above $T_{M}$, particles disperse. The free DNA strands in solution form a duplex below $T_{M}$. From the figure one can see that the transition from dispersed to aggregated phase is much sharper than the transition from single stranded DNA (ssDNA) to double stranded DNA (dsDNA) phase.

**Figure 5 micromachines-13-01102-f005:**
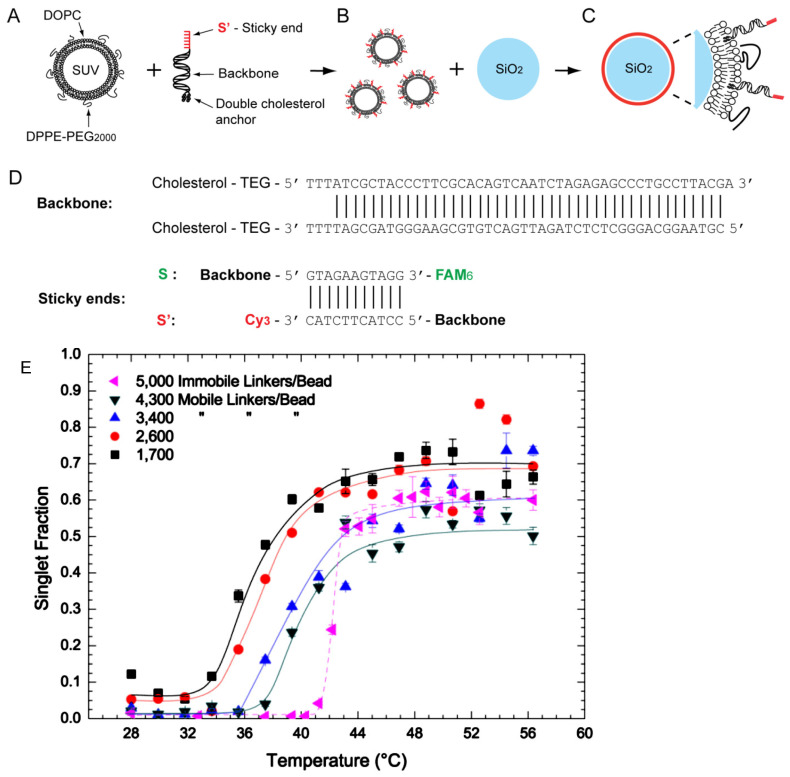
(**A**–**D**) represent the experimental scheme for creating particles grafted with mobile DNA strands. (**E**) shows temperature induced dissociation transition for different mobile-grafting densities and compares that with a system of particles grafted with immobile linkers. Reprinted with permission from [93]. Copyright 2013 American chemical society.

**Figure 6 micromachines-13-01102-f006:**
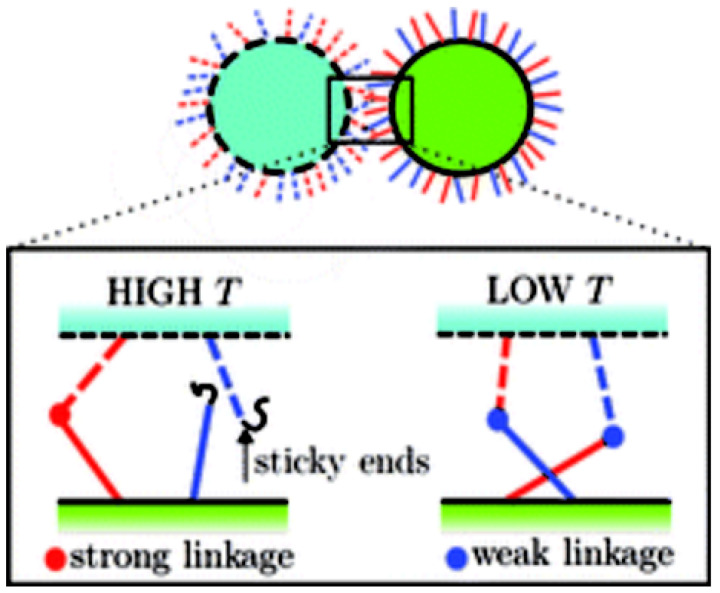
Reprinted with permission from Mongnetti et al. [111], shows the proposed scheme to introduce competing interactions. Copyright 2005 Royal society of Chemistry.

**Figure 7 micromachines-13-01102-f007:**
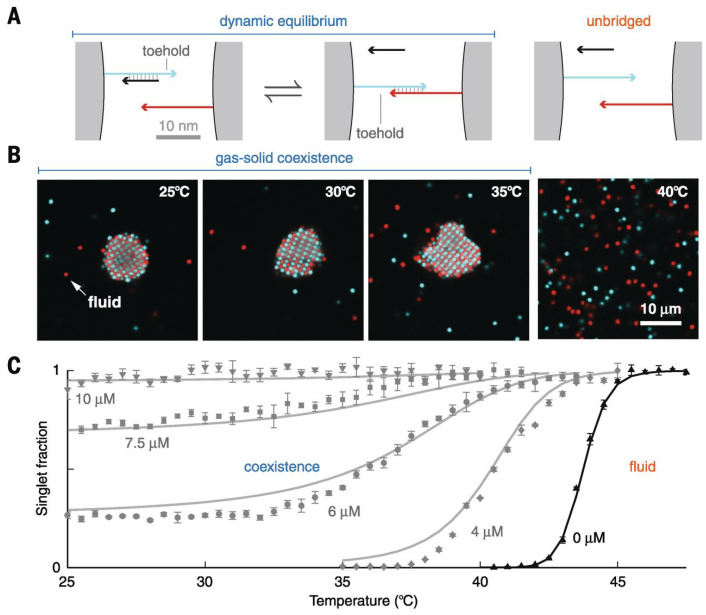
(**A**) shows the schematic representation of the strand displacement between the particles. (**B**) shows the confocal fluorescence micrograph of a binary suspension of DNA-grafted particles at different temperatures. (**C**) shows the singlet fraction for various concentrations of displacing strands; one can see the broadening of melting transition with increasing concentration of displacing strands. From [90], reprinted with permission from AAAS.

**Figure 8 micromachines-13-01102-f008:**
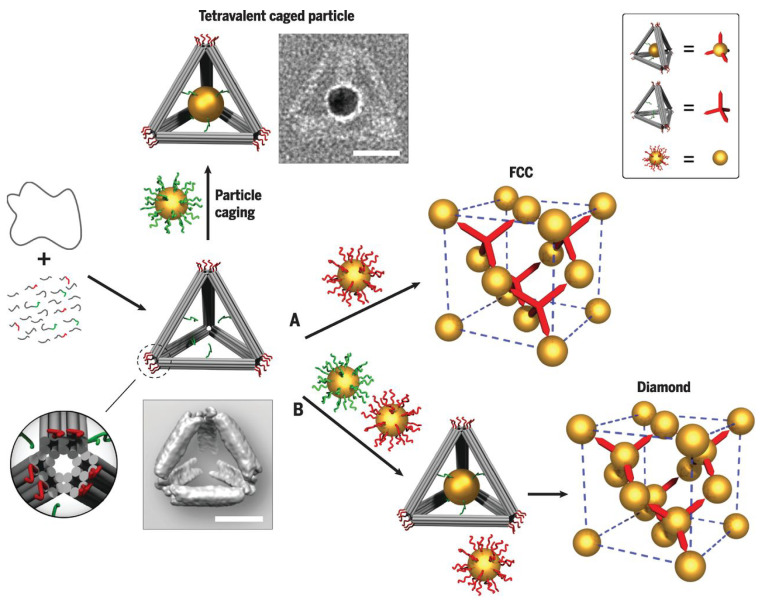
Illustration of how to generate tetrahedron origami cages with two sets of sticky ends: green one acting as an anchor to encapsulate and hold the nano-particles and the red ones acting as sticky patch to bind with basis particles. (**A**,**B**) represent two routes of assembly, resulting into FCC and diamond structures, respectively. Reprinted from [80], with permission from AAAS.

**Figure 9 micromachines-13-01102-f009:**
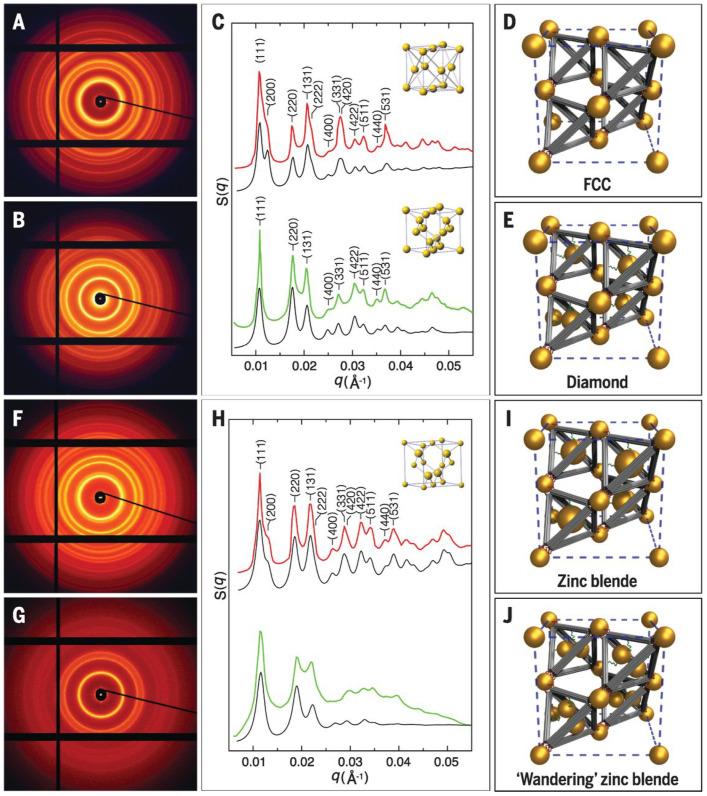
Structure of assemblies in a system shown in Figure 8. 2D SAXS pattern of the assembled structures formed by basis particles of core diameter 14.5 nm and, (**A**) tetrahedral origami cages, (**B**) trivalent cages. Upper and lower part of (**C**) shows the experimental (red and green, respectively), and calculated (black) SAXS 1D pattern corresponding to (**A**,**B**), respectively. (**D**,**E**) show the corresponding unit cell models. (**F**,**G**), again, show the 2D SAXS pattern formed by basis particles of core diameter 8.7 nm and (**F**) with tetravalent caged particles of diameter 8.7 nm and, (**G**) with guest pairs of diameter 8.7 nm, caged inside a tetrahedra. (**H**) corresponds to (**C**), and (**I**,**J**) show unit cell models for (**F**,**G**), respectively. Reprinted from [80], with permission from AAAS.

## Data Availability

Not applicable.

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
