# Peer review of "Self-Assembly of DNA-Grafted Colloids: A Review of Challenges"

_micromachines, 2022, doi:10.3390/mi13071102_

Round 1

Reviewer 1 Report

This short review describes the DNA-mediated self-assembly of colloids as a tool to assemble the materials of prescribed structure and properties. The DNA is useful building block for the programming the assembly of a desired structure, due to its highly directional and sequence specific binding properties.

Since the approach described in this manuscript are a useful for studies on micro/nano-scaled structures, materials, devices and systems, I consider the paper acceptable for publication at Micromachines after addressing the comments and thorough revision of the manuscript.

Comments

The text is properly discussed with citing the literature, but it is unfriendly for the reader to understand the specific examples because there are no figures including the molecular structure and schematic diagram. I would like you to add figures as appropriate to help the reader understand.

Reviewer 2 Report

The review is about DNA-coated colloidal particles, valuable building blocks for programmable colloidal assembly, with a specific focus on the current challenges in the field. Considering the importance and the success of the DNA self-assembly a review focusing on the challenges is useful, especially for researchers newly entering the field. The authors well summarize the current challenges and limitations, however; the review needs to incorporate the following aspects before being accepted for publication:

1) The manuscript does not discuss the literature on DNA-coated emulsions, and their use in sequential assembly. The authors do not refer to or mention these studies in their text. Examples include i) "Sequential self-assembly of DNA functionalized droplets." Nature communications 8, no. 1 (2017): 1-7. ii) "DNA-coated functional oil droplets." Langmuir 34, no. 34 (2018): 10073-10080.

2)      The authors should incorporate a figure or two summary figures from various literature highlighting various assembled structures.

3)      The manuscript mentions the advantages of having mobile DNA compared to fixed DNA. It would be useful to highlight the advantage and challenges of different strategies – mobile DNA, fixed DNA, and DNA on droplets.

4)      The paragraph describing the summary and future directions is important for any review. Currently, the paragraph summarizes the challenges in the field. However, lacks a detailed discussion on outlook or future directions. The section predominantly re-iterates the challenges rather than offering concrete suggestive solutions. A future directions paragraph with more specific prospective strategies or targeted directions toward overcoming the current challenges will make the manuscript more impactful.

Reviewer 3 Report

The authors introduced several methods to achieve self-assembled structures using DNA. They briefly discussed the control parameters to tune the effective interparticle interactions. The authors also summarized the challenges with DNA-grafted particles to generate a wide spectrum of structures. The topic itself is very interesting, however, the current manuscript is more of a summary report as opposed to an in-depth review. Figures that discuss the state-of-the-art progress, advantages and disadvantages of current methods, and tables with the summary of important parameters of the DNA-mediated self-assembly are required. Unless the authors substantially improve the quality of this manuscript, the reviewer cannot have a more positive opinion.

Round 2

Reviewer 2 Report

The authors have addressed the comments/concerns raised by the reviewer. With the inclusion of the new figures, references, and text, the article is ready to be accepted for publication.

Reviewer 3 Report

 The authors have revised the manuscript according to the reviewer's suggestion. Right now the quality of this manuscript has been improved. The reviewer recommends the publication of this review paper.